# Central American Parents’ Preferences for Content and Modality for a Family-Centered Intervention to Promote Healthful Energy Balance-Related Behaviors of Their Preschool-Age Children

**DOI:** 10.3390/ijerph19095080

**Published:** 2022-04-21

**Authors:** Elizabeth N. Díaz, Qun Le, Daniel Campos, Jesnny M. Reyes, Julie A. Wright, Mary L. Greaney, Ana Cristina Lindsay

**Affiliations:** 1Department of Exercise and Health Sciences, College of Nursing and Health Sciences, University of Massachusetts Boston, Boston, MA 02125, USA; elizabethdiaz9001@gmail.com (E.N.D.); d97.campos@gmail.com (D.C.); jesnnyr@icloud.com (J.M.R.); julie.wright@umb.edu (J.A.W.); 2Department of Public Health, Zuckerberg College of Health Sciences, University of Massachusetts Lowell, Lowell, MA 01854, USA; qun.le@uml.edu; 3Department of Health Studies, College of Health Sciences, University of Rhode Island, Kingston, RI 02881, USA; mgreaney@uri.edu

**Keywords:** energy balance-related behaviors, parents, Central American, immigrant, preschool, obesity, intervention

## Abstract

This formative research used a cross-sectional survey to assess interest in informational content and intervention modalities for the design of an early childhood obesity prevention intervention for Central American families from the Northern Triangle countries (El Salvador, Guatemala, and Honduras) living in the United States. A total of 74 parents (36 mothers, 38 fathers) with a mean age of 31.6 years (SD = 5.6) completed the survey; 59.5% of whom were born outside of the United States. Although most parents reported being very interested in receiving information related to the seven assessed energy balance-related behaviors, there were significant differences by parents’ gender and nativity. Most parents endorsed remote modalities for content delivery via text/SMS, WhatsApp, and e-mail. However, respondents were also receptive to in-person delivery provided by community health workers. There were also significant differences in parents’ preferences for intervention modalities by parents’ gender and nativity. Future steps should include investigating different intervention modalities and their integration into a linguistic and culturally sensitive family-based intervention to promote healthful energy balance-related behaviors of preschool-age children in Central American families originating from the Northern Triangle countries.

## 1. Introduction

Childhood obesity remains a significant public health problem in the United States and data indicate that Latinx children have one of the highest rates of childhood obesity [1,2]. Findings from a recent analysis of the National Health and Nutrition Survey (NHANES) data documented a sharp increase in overweight (26%) and obesity (13.8% class I obesity—body mass index > 95th percentile) among Latinx preschool-age (2–5 years) children compared to findings from previous cycles of the NHANES data [1,3]. These findings underscore the need for early obesity prevention efforts to reduce young children’s risk for obesity [4,5].

Parents are central to helping their children develop and maintain early healthful eating, physical activity (PA), and sedentary habits, and ultimately prevent childhood overweight and obesity [6,7,8]. Reaching parents and delivering key health messages during early childhood, when children’s early health habits are forming, is an ideal time to enable the development of healthful energy balance-related behaviors (EBRBs) and prevent child obesity [9,10].

The 5-2-1-0 Healthy Choices Count! mnemonic is designed to capture key recommendations to promote healthful EBRBs, including healthy eating (i.e., increase consumption of fruits and vegetables and water, and decrease consumption of unhealthy foods and sugar-sweetened beverages [SSBs]), increase PA, and decrease screen time [11,12]. The 5-2-1-0 Healthy Choices Count! message promotes five or more fruits and vegetables per day, 2 hours of screen time or less per day, 1 hour or more of PA per day, and zero SSBs per day, and is a nationally recognized evidence-based prevention framework to promote healthy habits and weight status in the United States. It has been endorsed by public health and medical organizations such as the Centers for Disease Control and Prevention and the American Academy of Pediatrics (AAP) [11,12]. The 5-2-1-0 Healthy Choices Count! was recently expanded to include promoting adequate sleep (≥10 hours sleep/night) [13].

Latinx children represent a fast-growing population in the United States and are expected to represent 31.9% of the total United States children population [14]. Nonetheless, to date, most childhood obesity interventions designed for Latinx families in the United States have focused on mothers and Mexican American and Mexican immigrant populations [15,16,17,18,19,20,21]. Central American immigrants from El Salvador, Guatemala, and Honduras, also known as the Northern Triangle countries, are the fastest-growing Latinx population in the United States [22]. Yet, there is a dearth of early childhood obesity prevention research designed to address the specific needs of Central American parents and children from the Northern Triangle countries living in the United States [15,16,17,18,19,20,21]. Although most Latinx groups share many cultural similarities, intervention efforts need to recognize and address the unique differences of the various Latinx groups to meet their specific needs [15,16,23]. Furthermore, there is a lack of obesity prevention research involving fathers, even though understanding paternal preferences for intervention design is essential to ensure fathers’ involvement in family-centered interventions [17,18,19,20,21]. Therefore, the purpose of this formative research was to assess the preferences for informational content, intervention modalities, language and use and access of communication technology of Central American mothers and fathers of preschool-age children (2–5 years) from the Northern Triangle countries to develop a family-centered intervention to promote healthful EBRBs and prevent childhood obesity.

## 2. Materials and Methods

Guided by the Family Ecological Model (FEM) [24], this cross-sectional study was conducted as part of formative research to inform the design of a family-centered early childhood obesity prevention intervention for Central American families living in the United States with at least one parent who self-identifies as Central American from El Salvador, Guatemala, and Honduras, also known as the Northern Triangle countries, with at least one child between 2 and 5 years of age [25,26]. Parents were eligible to participate if they: (a) self-identified as Central American from the three Northern Triangle countries—Guatemala, El Salvador, or Honduras; (b) had at least one child between the ages of 2 and 5 years; (c) were 21 years of age or older; (d) lived in Massachusetts; (e) spoke Spanish or English; and (f) provided verbal informed consent.

### 2.1. Data Collection

Data collection occurred between April and August 2020. Participants were recruited through flyers posted on social media (e.g., Facebook and WhatsApp). Participants were also recruited through a network or snowball sampling approach, with study participants asking their Central American friends if they would be interested in participating in this study [27]. Interested individuals called the telephone number included on the flyer and were screened for study eligibility via telephone by study staff. After determining eligibility and before study enrollment, eligible participants were read the informed consent form in their preferred language (Spanish or English) by a trained bilingual, bicultural interviewer. After providing verbal informed consent, participants completed an interviewer-administered survey in their preferred language (Spanish or English) via telephone [28].

In brief, the survey was based on the public health and obesity prevention goals put forth in 5-2-1-0 + 10 message [29,30] and the research teams’ previous qualitative research with Latinx parents living in the United States [31,32,33,34]. The survey was first developed in English and then professionally translated into Spanish, pilot tested with three Central American parents, and revised as needed. Data from the pilot interviews are not included in this paper, which focuses on participants’ responses to survey questions assessing preferences for informational content (7 items), intervention modalities (11 items), language preference (1 item), and communication access (5 items) and use (3 items) for family-based intervention to promote healthful EBRBs and prevent obesity among preschool-age children.

### 2.2. Survey Measures

#### 2.2.1. Interest in Informational Content

The following questions measured respondents’ interest in receiving information about health EBRBs: “The next questions are about how much you would be interested in learning more about the following topics related to your child’s preschool-age health: Helping your child: (1) eat more healthy foods such as fruits and vegetables, etc.; (2) eat less unhealthy or “junk” (e.g., chocolates, candy, etc.) and fast food (e.g., hamburgers, fries, etc.), (3) drink less sugar-sweetened beverages (e.g., soft drinks, artificial juices, etc.), (4) drink more water, (5) be more physically active, (6) reduce the use of electronics (e.g., television, iPad, iPhone, computer, video game, etc.) or having less screen-time, (7) having adequate night sleep (e.g., ≥10 hours/night).” Response options were on a 5-point Likert scale: 1 = not interested to 5 = very interested.

#### 2.2.2. Preference for Intervention Modality

The following questions assessed parents’ preferences for different intervention modalities for delivery of information: “If you were to enroll in a health promotion and obesity prevention program for parents of preschool-age children, what would be your preference for the delivery of such information? (1) e-mail, (2) text or SMS, (3) WhatsApp, (4) social media (e.g., Facebook, Instagram, Pinterest, etc.), (5) English language website, (6) Spanish language website, (7) telephone calls by trained community health workers, (8) individual sessions (6–8 weeks) delivered by trained community health workers, (9) short duration courses (6–8 weeks) offered by trained community health workers, (10) short-duration courses (6–8 weeks) offered by trained parents like me or peer parents, and (11) printed materials. Response options were on a 5-point Likert scale: 1 = completely disagree–5 = completely agree.

#### 2.2.3. Language Preference and Technology Access and Use

The following question measured language preference. “If you were to enroll in a health promotion program for parents of preschool-age children what would be your language preference to receive information?” (Spanish, Either Spanish or English, English). Additionally, the following questions measured technology access and use: Do you have access to a computer at home? (Yes, No). Do you have access to the Internet at home? (Yes, No). How often do you check your e-mail? (Daily, More than once a week, Less than once a week, Don’t use e-mail). Do you have a mobile telephone where you can receive text messages? (Yes, No). Do you use WhatsApp? (Yes, No). If yes, how often do you check your WhatsApp messages? (Daily, More than once a week, Less than once a week). Do you use social media (e.g., Facebook, Instagram, Pinterest)? (Yes, No). If yes, how often do you check your social media? (Daily, More than once a week, Less than once a week).

#### 2.2.4. Socio-Demographics, Acculturation, and Parent and Child Reported Weight Status

Participants reported their age (open-ended in years), marital status (Married, Live together with partner, Separated or Divorced, Never married, Widower), education attainment level (Less than high school diploma, High school diploma, More than high school diploma (e.g., Associate Degree, College or University)), family annual income before paying taxes (<US$30,000; ≥US$30,000–<US$40,000; ≥US$40,000–<US$50,000; ≥US$50,000–<US$60,000; ≥US$60,000–<US$70,000; ≥US$70,000–<US$80,000; ≥US$80,000), health insurance status (Yes, No), their perception of their weight status (Underweight, Normal weight, Overweight, Obese), and whether a doctor ever expressed health concerns about their weight (Yes, No). In addition, participants reported their country of birth (El Salvador, Guatemala, Honduras, United States, Other, please specify) and the number of years living in the United States if foreign-born (open ended in years).

Parents also reported their perceived weight status, completed an acculturation scale, and their child’s socio-demographic characteristics and weight status using measures used in our previous studies [34,35,36]. When answering these survey questions, parents with more than one preschool-age child were asked to think of their oldest age-eligible child. Participants’ acculturation level was assessed using the Short Acculturation Scale for Hispanics (SASH), a 12-item scale validated for use with Latinx populations [37]. The SASH has 12 items that assess language use, media use, and ethnic social relations. The scale has good reliability (Cronbach’s alpha reliabilities 0.92–0.89 for the overall SASH scale, 0.89 for language use, 0.88 for media preference, and 0.72 for ethnic and social relations [37,38]. Acculturation scores were computed by averaging across the 12 items, measured on a scale of 1 to 5, and scores were then dichotomized (high vs. low). The scale developers recommend an average of 2.99 as the cut-point scores equal to or above this point represent higher levels of acculturation, and scores below this point represent lower levels of acculturation [36]. We used the recommended cut-points to categorize respondents as having a low acculturation level (SASH < 2.99) or a high acculturation level (SASH ≥ 2.99) [37].

The average time for completing the entire survey was 15 minutes. Participants received a $25 gift card at the end of the interview for their participation. The Institutional Review Board (IRB #2020086) at the University of Massachusetts Boston approved the study protocol, and all participants provided informed consent.

### 2.3. Data Analysis

Means and standard deviations were calculated for all continuous variables and frequencies and percent for categorical variables. Based on the data distribution, response options for interest in informational content, intervention modality, preferred language, and country of nativity were dichotomized for analysis. Chi-square tests, and Fisher’s exact tests were used as appropriate to determine if there were differences between mothers and fathers in socio-demographic characteristics (<US$40,000 vs. ≥US$40,000; <high school diploma vs. high school diploma; married/cohabitating vs. divorced/separated/single; US-born vs. foreign-born), acculturation level (SASH ≥ 2.99 vs. SASH < 2.99), parent and child reported weight status (normal weight vs. overweight/obese). Chi-square tests, and Fisher’s exact tests were also conducted to determine if there were differences between mothers’ and fathers’ interest in informational content related to EBRBs (Very interested vs. Interested/Neutral), preference for intervention modalities (Completely agree vs. Neutral/Disagree), and language preference (Spanish vs. Spanish/English or English). Similar analyses were conducted to determine if there were differences in parents’ access to communication technology (Yes vs. No) and use (Yes vs. No), and frequency (Daily vs. More than once a week, Less than once a week). All analyses were performed using SAS 7.1 (SAS Institute, Cary, NC, USA).

## 3. Results

### 3.1. Participants Characteristics

Seventy-four Central American parents (36 mothers, 38 fathers) participated in this study. Parents’ mean age was 31.6 years (SD = 5.6), more than half had a high school education or less, and approximately 40% were classified as having a low income (<US$40,000/year). Additionally, more than half of repondents (59.5%) were foreign-born, and of these, 56.8% were born in El Salvador, 22% in Honduras, and 20.9% in Guatemala. Foreign-born participants had lived in the United States for an average of 12.9 years (SD = 7.1), and all foreign-born respondents were classified as having low acculturation levels (<2.99 SASH score). Moreover, approximately one-third of parents perceived themselves as being overweight (32.4%). There were no significant differences by parents’ gender or country of nativity in self-reported weight status.

Regarding the children respondents reported on when completing the survey, 53.8% were female, 87.8% were born in the United States. In addition, 12.2% of parents perceived their preschool-age child as being overweight. Additional information on socio-demographic and cultural variables are presented in Table 1.

### 3.2. Interest in Informational Content

Overall, most parents reported being very interested in receiving information focused on all assessed EBRBs (Table 2). In terms of healthy eating behaviors, approximately 62%, 58%, 50%, and 47.3% of respondents reported being very interested in receiving content related to increasing water consumption, promoting consumption of fruits and vegetables, limiting consumption of unhealthy or “junk” foods, and limiting consumption of SSBs, respectively (Table 2). A greater proportion of mothers than fathers reported being very interested in receiving informational content about increasing water consumption (77.8% vs. 47.4%, *p* = 0.01), promoting consumption of ≥5 fruits and vegetables/day (75% vs. 42.1%; *p* = 0.004), limiting consumption of unhealthy or “junk” foods (66.7% vs. 34.2%, *p* = 0.002), and limiting consumption of SSBs (63.9% vs. 31.6%, *p* = 0.003). Furthermore, more foreign-born than US-born parents reported being very interested in content related to increasing water consumption (75% vs. 43.3%, *p* = 0.003) and limiting consumption of SSBs (56.8% vs. 33.3%, *p* = 0.04).

In terms of the healthy 24-hour movement behaviors, approximately 58%, 56.8%, and 47.3% reported being very interested in content related to promoting adequate night sleep, limiting screen time, and promoting ≥1 hour PA/day, respectively. Similar to the preference for content on healthy eating behaviors, more mothers than fathers reported being very interested in the assessed 24 h movement behaviors (Table 2). However, these differences were significant only for interest in content promoting ≥1 hour or more of PA/day (67.7% vs. 29%, *p* = 0.007) and limiting screen time to ≤2 hour/day (67.7% vs. 47.4%, *p* = 0.03). Furthermore, a greater proportion of foreign-born than US-born parents reported being very interested in all three healthy 24-hour movement behaviors, but the only significant difference was for content about limiting screen time to ≤2 hour/day (65.9% vs. 43.3%, *p* = 0.04).

### 3.3. Preferred Intervention Modalities and Language

Most parents preferred that intervention content be delivered remotely, with 81% delivering via text/SMS messages, 73% via WhatsApp, and 66.2% via e-mail (Table 3). A greater proportion of US-born than foreign-born parents (83.3% vs. 54.5%, *p* = 0.01) supported having content delivered via e-mail. On the other hand, more foreign-born than US-born parents preferred content be delivered via Spanish-language websites (72.7% vs. 43.3%, *p* = 0.01). Although more mothers than fathers endorsed content delivery via text/SMS and WhatsApp, these differences were not significant (Table 3).

Parents also were receptive to in-person delivery of information. Approximately 70% and 66.2% of parents endorsed group and individual sessions lead by trained community health workers (CHWs), respectively. Additionally, 60.8% supported information being delivered via telephone calls by CHWs. More mothers than fathers endorsed group sessions lead by CHWs (86.1% vs. 55.3%, *p* = 0.001) and peer-parents (86.1% vs. 50%, *p* = 0.004), as well as individual sessions with CHWs (80.6% vs. 52.6%, *p* = 0.01) (Table 3). A greater percentage of foreign-born than US-born parents endorsed CHWs delivering content via group session (79.5% vs. 56.7%, *p* = 0.04) and telephone calls by CHWs (70.5% vs. 46.7%, *p* = 0.04) (Table 3). Furthermore, 67.6% of repondents endorsed content delivery via group sessions lead by peer parents. Finally, most parents preferred that information be delivered in Spanish (59.5%), with nearly all foreign-born respondents (95.5% vs. 6.7%, *p* < 0.0001) indicating a preference for Spanish.

### 3.4. Communication Technology Access and Use

All respondents reported having access to a mobile phone to receive text messages (Table 4). The majority (93.2%) reported using social media; 91.9% reported using WhatsApp, and 89.2% reported checking WhatsApp/text messages/social media daily. A greater proportion of mothers than fathers reported using WhatsApp (100% vs. 84.2%, *p* = 0.01), social media (100% vs. 86.8%, *p* = 0.03), and checking these daily (94.4% vs. 76.3%, *p* = 0.03). Moreover, a greater proportion of foreign-born than US-born parents reported using WhatsApp (100% vs. 80%, *p* = 0.002), and checking messages daily (95.5% vs. 80%, *p* = 0.04). In addition, all respondents reported having internet access at home, and approximately 81% reported having access to a computer at home. A greater proportion of US-born than foreign-born parents (100% vs. 68.2%, *p* < 0.001) reported having access to a home computer. Additionally, only 40.5% of respondents reported checking their e-mail daily, with a greater proportion of US-born than foreign-born parents reporting they checked e-mail daily (56.6% vs. 22.7%, *p* < 0.001) (Table 4).

## 4. Discussion

This paper describes the findings of formative research assessing Central American parents’ interest in informational content, preference for language and communication channels for delivery of information, and access and use of communication technology. This information will inform the development of an intervention to promote healthful EBRBs among preschool-aged children (2–5 years) whose parents self-identify as Central Americans from the Northern Triangle (El Salvador, Guatemala, and Honduras) countries. Although most parents reported being very interested in receiving information related to all assessed health-promoting EBRBs, there were significant differences by parents’ gender and nativity. Specifically, more mothers than fathers reported being very interested in six of the seven assessed EBRBs. Additionally, a greater proportion of foreign-born than US-born parents reported being very interested in limiting consumption of SSBs, increasing water consumption, and limiting screen time. This information has implications when designing interventions to promote early EBRBs and prevent obesity among preschool-aged children (2–5 years) in Central American families from the Northern Triangle countries [39,40,41]. These findings are supported by previous research and suggest the importance of tailoring information by parent gender and nativity [41,42,43,44,45].

Overall, most parents participating in this study endorsed delivering content using remote communication channels, including text/SMS, WhatsApp, and e-mail. In-person delivery by CHWs was the second most endorsed intervention modality. Prior research with Latinx populations suggests the feasibility of technology-enhanced intervention delivery [45,46,47] and in-person interventions delivered by CHWs [48,49,50]. Nonetheless, this study identified significant differences in preferred modalities between mothers and fathers, and foreign-born and US-born parents and should be considered when designing interventions to prevent child obesity and promote overall health of preschool-aged (2–5 years) children in Central American families in the United States. This information is important and has implications for the design of future interventions, suggesting the need for tailoring by parents’ gender and nativity. Nonetheless, these differences warrant further exploration with a broader sample of parents who self-identify as Central Americans from the Northern Triangle countries living in the United States. As promoting healthful behavior is complex, future interventions will likely need to include multiple components and modalities as a single delivery channel will likely be insufficient to effectively deliver family-centered interventions targeting EBRBs tailored to meet the needs of foreign-born and US-born Central American from the Northern Triangle in the United States [23,51,52].

Finally, it should be noted that approximately one-third of the participating parents reported being overweight (32.4%), and 12.2% perceived their preschool-age child (2–5 years) as being overweight. A recent prospective analysis of data from a randomized controlled trial conducted in Tennessee with a sample that was 91% Hispanic found that child age, parent body mass index, and child overweight status were significantly associated with the odds of child obesity [49]. Supported by previous research, our findings have important implications and suggest the importance of family-centered interventions addressing lifestyle behaviors of both parents and children to reduce the risk of early childhood obesity [23,51,52,53].

Limitations of the current research should be considered. First, as is often the case with formative research, the sample is a small, convenience sample of parents of preschool-aged children (2–5 years) from the Northern Triangle living in Massachusetts, which limits generalizability and ability to conduct additional analysis [25,26]. Participants may have heightened interest and awareness regarding the study’s topics. Moreover, snowball sampling and the fact that approximately one-third (*n* = 24) of parents were from the same household may have resulted in parents who share similar views enrolling in this study [54]. Thus, further research is needed to increase generalizability and to explore whether results apply to a broader group of Central American parents from the Northern Triangle countries living in the United States. Finally, our study relied on self-report for perceived weight status of both parents and preschool-age children (2–5 years). Therefore, future studies should consider including measured weight and height of parents and children.

## 5. Conclusions

This study provides important information that can help guide the development of family-centered interventions to promote healthful EBRBs of preschool-aged children (2–5 years) in Central American families from the Northern Triangle countries. Study findings suggest that future steps for intervention design should investigate the different intervention modalities and their integration into a pilot testing of a linguistic and culturally sensitive family-centered intervention to promote healthful EBRBs of preschool-age children (2–5 years) in Central American families with both parents and evaluate differences in engagement and interaction with study materials and modalities between mothers and fathers and foreign-born and US-born parents.

## Figures and Tables

**Table 1 ijerph-19-05080-t001:** Socio-demographic and acculturation characteristics of study participants (*n* = 74).

Parents’ Characteristics	Mothers *n* = 36 (%)	Fathers *n* = 38 (%)	*p*-Value	Total Sample *n* = 74 (%)
**Socio-demographic variables**				
**Age (mean SD)**	32.5 (5.5)	30.7 (5.7)	0.16	31.6 (5.6)
**Marital status**				
Married/Cohabitating	29 (80.6)	31 (81.6)	0.91	60 (81.1)
Divorced/Separated/Single	7 (19.4)	7 (18.4)		14 (18.9)
**Educational attainment**				
≤High school diploma	20 (56.6)	27 (71.0)	0.17	47 (63.5)
High school diploma	16 (44.4)	11 (29.0)		27 (36.5)
**Household annual income**				
<US$40,000	15 (41.7)	20 (52.6)	0.56	35 (47.3)
≥US$40,000	16 (44.4)	16 (42.1)		32 (43.2)
Declined to report/Don’t Know	5 (13.9)	2 (5.3)		7 (9.5)
**Acculturation variables**				
**Nativity**				
US-born	16 (44.4)	14 (36.8)	0.51	30 (40.5)
Foreign-born	20 (55.6)	24 (63.2)		44 (59.5)
**Parents’ country of origin ^1^**				
El Salvador	13 (65.0)	12 (50.0)		25 (56.8)
Honduras	5 (25.0)	5 (20.8)		10 (22.7)
Guatemala	2 (10.0)	7 (29.2)		9 (20.5)
**Years living in the United States ^1^**	15.2 (7.2)	11.3 (6.7)	0.07	12.9 (7.1)
**SASH score ^2^**				
<2.99	20 (55.6)	25 (65.8)	0.37	45 (60.8)
≥2.99	16 (44.4)	13 (34.2)		29 (39.2)
**Healthcare variables**				
**Health insurance (yes)**	36 (100)	38 (100)		74 (100)
**Child enrolled in the WIC program**				
Yes	29 (80.6)	27 (71.1)	0.34	56 (75.7)
No	7 (19.4)	11 (28.9)		18 (24.3)
**Parent weight-related variables**				
**Parent self-reported weight status**				
Normal weight	1 (2.8)	4 (10.5)	0.09	5 (6.8)
Overweight	19 (52.8)	26 (68.4)		45 (60.8)
Obese	16 (44.4)	8 (21.1)		24 (32.4)
**Doctor expressed concerns about parent’s weight**				
Yes	16 (44.4)	10 (26.3)	0.10	26 (35.1)
No	20 (50.6)	28 (73.7)		48 (64.9)
**Child characteristics**				
**Age (mean; SD)**	3.1 (1.2)	3.2 (1.8)	0.89	3.5 (1.2)
**Sex**				
Male	15 (41.7)	16 (42.1)	0.96	31 (46.2)
Female	21 (58.3)	22 (57.9)		43 (53.8)
**Place of nativity**				
United States	33 (91.7)	32 (84.2)	0.33	65 (87.8)
Other	3 (8.3)	6 (5.8)		9 (12.2)
**Parent perceived child weight status**				
Underweight	10 (27.8)	5 (13.2)	0.67	15 (20.3)
Normal weight	21 (58.3)	29 (76.3)		50 (67.6)
Overweight	5 (13.9)	4 (10.5)		9 (12.2)

^1^ Only foreign-born parents. ^2^ All parents.

**Table 2 ijerph-19-05080-t002:** Participants’ preferences for informational content to promote healthful energy balance-related behaviors of their preschool-aged children.

Energy Balance-Related Behaviors	Mothers *n* = 36 (%)	Fathers *n* = 38 (%)	*p*-Value	Foreign-Born *n* = 44 (%)	US-Born *n* = 30 (%)	*p*-Value	Total Sample *n* = 74 (%)
**Promotion of Healthy Dietary Behaviors**							
**Limiting consumption unhealthy/junk foods**							
Very Interested	24 (66.7)	13 (34.2)	0.002	22 (50.0)	15 (50.0)	0.92	37 (50.0)
Interested/Neutral	12 (33.3)	25 (65.8)		22 (50.0)	15 (50.0)		37 (50.0)
**Limiting consumption SSBs**							
Very Interested	23 (63.9)	12 (31.6)	0.003	25 (56.8)	10 (33.3)	0.04	35 (47.3)
Interested/Neutral	13 (36.1)	26 (68.4)		19 (43.2)	20 (66.7)		39 (52.7)
**Consuming ≥ 5 fruits and vegetables/day**							
Very Interested	27 (75.0)	16 (42.1)	0.004	27 (61.4)	16 (53.3)	0.77	43 (58.1)
Interested/Neutral	9 (25.0)	22 (57.9)		17 (38.6)	14 (46.7)		31 (41.9)
**Increasing water consumption**							
Very Interested	28 (77.8)	18 (47.4)	0.01	33 (75.0)	13 (43.3)	0.003	46 (62.2)
Interested/Neutral	8 (22.2)	20 (52.6)		11 (25.0)	17 (56.7)		28 (37.8)
**Promotion of Healthy 24-hour Movement Behaviors**							
**Limiting screen time to ≤2 h/day**							
Very Interested	24 (66.7)	18 (47.4)	0.03	29 (65.9)	13 (43.3)	0.04	42 (56.8)
Interested/Neutral	12 (33.3)	20 (52.6)		15 (34.1)	11 (56.7)		32 (43.2)
**Promoting ≥ 1 hour/day of PA**							
Very Interested	24 (66.7)	11 (29.0)	0.007	23 (52.3)	12 (40.0)	0.24	35 (47.3)
Interested/Neutral	12 (33.3)	27 (71.0)		21 (47.7)	18 (60.0)		31 (52.7)
**Promoting ≥ 10 hour sleep/night**							
Very Interested	24 (66.7)	19 (50.0)	0.11	29 (65.9)	14 (46.7)	0.13	43 (58.1)
Interested/Neutral	12 (33.3)	19 (50.0)		15 (34.1)	16 (53.3)		31 (41.9)

**Table 3 ijerph-19-05080-t003:** Participants’ preferences for intervention modality for a family-based intervention to promote healthful energy balance-related behaviors of their preschool-aged children (2–5 years).

Preferred Intervention Modalities	Mothers *n* = 36 (%)	Fathers *n* = 38 (%)	*p*-Value	Foreign-Born *n* = 44 (%)	US-Born *n* = 30 (%)	*p*-Value	Total Sample *n* = 74 (%)
**In-Person Modalities**							
**Group sessions by peer-parents**							
Completely agree	31 (86.1)	19 (50.0)	0.001	30 (68.2)	20 (66.7)	0.89	50 (67.6)
Neutral/Disagree	5 (13.9)	19 (50.0)		14 (31.8)	10 (33.3)		24 (32.4)
**Group sessions by CHWs ^1^**							
Completely agree	31 (86.1)	21 (55.3)	0.004	35 (79.5)	17 (56.7)	0.04	52 (70.3)
Neutral/Disagree	5 (13.9)	17 (36.8)		9 (20.5)	13 (43.3)		22 (29.7)
**Individual sessions by CHWs**							
Completely agree	29 (80.6)	20 (52.6)	0.01	29 (65.9)	20 (66.7)	0.95	49 (66.2)
Neutral/Disagree	7 (19.4)	18 (29.0)		15 (34.1)	10 (33.3)		25 (33.8)
**Remote Modalities**							
**Telephone calls from CHWs**							
Completely agree	15 (27.7)	20 (52.6)	0.34	31 (70.5)	14 (46.7)	0.04	49 (66.2)
Neutral/Disagree	21 (58.3)	18 (47.4)		13 (29.5)	16 (53.3)		25 (33.8)
**E-mail**							
Completely agree	26 (94.4)	23 (60.5)	0.001	24 (54.5)	25 (83.3)	0.01	49 (66.2)
Neutral/Disagree	10 (5.6)	15 (29.5)		20 (45.5)	5 (16.7)		25 (33.8)
**WhatsApp**							
Completely agree	28 (77.8)	26 (68.4)	0.37	34 (77.3)	20 (66.7)	0.32	54 (73.0)
Neutral/Disagree	8 (13.9)	12 (23.7)		10 (22.7)	10 (33.3)		20 (27.0)
Text/SMS							
Completely agree	33 (91.7)	27 (71.1)	0.04	34 (77.3)	26 (86.7)	0.31	60 (81.1)
Neutral/Disagree	3 (8.3)	11 (28.9)		10 (22.7)	4 (13.3)		14 (18.9)
**Social media (Facebook)**							
Completely agree	21 (58.3)	8 (21.1)	0.001	18 (40.9)	11 (36.7)	0.72	29 (39.2)
Neutral/Disagree	15 (22.2)	30 (39.5)		26 (59.1)	19 (63.3)		45 (60.8)
**Spanish-language website**							
Completely agree	24 (66.7)	21 (55.3)	0.32	32 (72.7)	13 (43.3)	0.01	49 (66.2)
Neutral/Disagree	12 (33.3)	17 (44.7)		12 (27.3)	17 (56.7)		25 (33.8)
**English-language website**							
Completely agree	20 (55.6)	7 (18.4)	0.001	14 (31.8)	13 (43.3)	0.31	27 (36.4)
Neutral/Disagree	16 (33.3)	31 (81.6)		30 (68.2)	17 (56.7)		47 (63.5)
**Printed materials**							
Completely agree	29 (80.6)	18 (47.4)	0.003	30 (68.2)	17 (56.7)	0.31	47 (63.5)
Neutral/Disagree	7 (13.9)	20 (36.8)		14 (31.8)	13 (43.3)		27 (36.4)

^1^ CHW—community health worker.

**Table 4 ijerph-19-05080-t004:** Participants’ language preferences for receipt of information and communication technology access and frequency of use.

Preferences and Access	Mothers *n* = 36 (%)	Fathers *n* = 38 (%)	*p*-Value	Foreign-Born *n* = 44 (%)	US-Born *n* = 30 (%)	*p*-Value	Total Sample *n* = 74 (%)
**Language preference ^1^**							
Spanish	22 (61.1)	22 (57.9)	0.78	42 (95.5)	2 (6.7)	<0.0001	44 (59.5)
Either Spanish or English	13 (36.1)	9 (23.7)		2 (4.5)	20 (56.7)		22 (29.7)
English	1 (2.7)	7 (18.4)		0	8 (36.6)		8 (10.8)
**Access to a computer at home**							
Yes	30 (83.3)	30 (78.9)	0.63	30 (68.2)	30 (100)	0.0007	60 (81.1)
No	6 (16.7)	8 (21.1)		14 (31.8)	0		14 (18.9)
**Access to internet at home**							
Yes	36 (100)	38 (100)	1.00	44 (100)	30 (100)	1.00	74 (100)
**Frequency check e-mail ^2^**							
Daily	26 (72.2)	20 (52.6)	0.08	10 (22.7)	20 (56.6)	0.0001	30 (40.5)
More than once a week	8 (22.2)	7 (18.4)		8 (18.2)	7 (23.4)		15 (20.3)
Less than once a week	2 (5.6)	11 (29.0)		26 (59.1)	3 (10.0)		29 (39.2)
**Access to mobile telephone (SMS/text)**							
Yes	36 (100.0)	38 (100)	1.00	36 (100)	38 (100)	1.00	74 (100)
**Access and use of WhatsApp**							
Yes	36 (100)	32 (84.2)	0.01	44 (100)	24 (80.0)	0.002	68 (91.9)
No	0	6 (15.8)		0	6 (20.0)		6 (8.1)
**Social media (e.g., Facebook, Instagram, Pinterest) use**							
Yes	36 (100)	33 (86.8)	0.03	42 (95.5)	27 (90.0)	0.36	69 (93.2)
No	0	5 (13.2)		2 (4.5)	3 (10.0)		5 (6.8)
**Frequency use of WhatsApp, SMS/text, and social media ^2^**							
Daily	34 (94.4)	29 (76.3)	0.03	42 (95.5)	24 (80.0)	0.04	66 (89.2)
More than once a week	2 (5.6)	8 (21.1)		0	6 (20.0)		6 (8.1)
Less than once a week	0	1 (2.6)		2 (4.5)	0		2 (2.7)

^1^*p*-value was calculated between “Spanish” vs. “Either Spanish or English + English”; ^2^
*p*-value was calculated between “daily” vs. “all other options”.

## Data Availability

Data are available upon reasonable requests submitted to the corresponding author.

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
