# Peer review of "Central American Parents’ Preferences for Content and Modality for a Family-Centered Intervention to Promote Healthful Energy Balance-Related Behaviors of Their Preschool-Age Children"

_ijerph, 2022, doi:10.3390/ijerph19095080_

Round 1

Reviewer 1 Report

Dear authors,

your paper is very good. You need to add more information to conclusion section based on your results and discussion. Which are the limitations of your study? 

Author Response

RESPONSE: As suggested by this reviewer have added information to the conclusion section based on the results and discussion. Please see page 11, lines 336-342. In addition, we have added information on the study’s limitations, in response to this reviewer’s question. Please see pages 10 and 11, lines 320-332.

Reviewer 2 Report

Minor points and suggestions

  • It is suggested that the age means and standard deviation of parents and also their role (number of mothers/fathers) be mentioned in the abstract.
  • The introduction did not well write about the research question and didn't cite previous reports in this area.
  • The authors have mentioned “A detailed description of the recruitment methods and data collection has been published elsewhere”. It is suggested that the authors mention this information here in this study.
  • It is suggested that the authors speak more in the discussion section regarding the results and write about future implications/applications of the results found in the research.

Author Response

It is suggested that the age means and standard deviation of parents and also their role (number of mothers/fathers) be mentioned in the abstract.

RESPONSE: We have added the mean age and standard deviation of parents and the number of mothers and fathers to the abstract. Please see the abstract:  the new text is in red font.

The introduction did not well write about the research question and didn't cite previous reports in this area.

RESPONSE: We have revised the text to identify the purpose of the study, which is the research questions, and have emphasized the lack of previous studies with this population addressing the study’s research question. Please see page 2, lines 60 and 61, 63-65, 68-69, and lines 71-74.

The authors have mentioned “A detailed description of the recruitment methods and data collection has been published elsewhere”. It is suggested that the authors mention this information here in this study.

RESPONSE: We have added the information requested by the reviewer to the methods section. Please see page 2, lines 82-86, and pages 2 and 3, lines 88-97.

It is suggested that the authors speak more in the discussion section regarding the results and write about future implications/applications of the results found in the research.

RESPONSE: We have added the information requested by this reviewer in several parts of the discussion section. Please see page 10, lines 289-293, 300-310, and pages 10 and 11, lines 316-319.

Reviewer 3 Report

Thank you very much for the opportunity to review an interesting article.

Here are some suggestions:

  • in the introduction, there is no theoretical basis for the proper diet of preschool children, there are only recommendations
  • in the methodological part, as the authors rightly noticed, they did not ask for data enabling BMI calculation, which is a big mistake of this study
  • I did not find information about the age of the children, which is important for establishing a proper diet
  • in the discussion there are no references to similar studies on the use of applications or programs for the prevention of healthy eating in preschool children

Author Response

In the introduction, there is no theoretical basis for the proper diet of preschool children, there are only recommendations.

RESPONSE: We respectfully disagree with the reviewer, this study is not about dietary assessment of preschool-age children (2-5 years of age), but about parents’ preferences for an intervention targeting recommendations for healthful behaviors of children according to the current recommendations in the United States for such behaviors, which we cite in the introduction section. Please see page 2, lines 46-56.

In the methodological part, as the authors rightly noticed, they did not ask for data enabling BMI calculation, which is a big mistake of this study

RESPONSE: We agree that this is a limitation of the study, and we indicate this in the limitations section. Please see page 11, lines 329-332 for the limitation of relying on self-report of weight status and not measuring weight and height and calculating BMI.

I did not find information about the age of the children, which is important for establishing a proper diet.

RESPONSE:  The age of the children is mentioned in several parts of the article including in the introduction (page 2, line 73), methods (page 2, lines 84 and 85). We use the language of “preschool-aged children” throughout the article. In the discussion, we have now added the age range when referring to preschool-aged children to make it more explicit the age range.  Please see discussion and conclusion sections, pages 10, lines 282, 291, 301-302, 312, and page 11, lines 322, 330, 335-336, and 340.

In the discussion, there are no references to similar studies on the use of applications or programs for the prevention of healthy eating in preschool children. 
RESPONSE: We respectfully disagree with the reviewer's comment. We cite several studies in the discussion section for comparing the findings of our study with that of prior studies. Please see the discussion section, pages 10 and 11, references cited are #23, and #39 to #52.

Round 2

Reviewer 2 Report

Hi all,

The authors have tried to improve the quality of the manuscript, and they did, so I think the paper is now good for publication in the Journal.

Regards,
ES,